# Packaging Communication as a Tool to Reduce Disgust with Insect-Based Foods: Effect of Informative and Visual Elements

**DOI:** 10.3390/foods12193606

**Published:** 2023-09-28

**Authors:** Natalia Naranjo-Guevara, Bastian Stroh, Sonja Floto-Stammen

**Affiliations:** Research Group Business Innovation, Fontys University of Applied Sciences, Tegelseweg 225, 5912 BG Venlo, The Netherlands; bastianstroh@gmail.com (B.S.); s.flotostammen@fontys.nl (S.F.-S.)

**Keywords:** consumer acceptance, edible insects, entomophagy, insect products, labels, marketing strategies, willingness to eat

## Abstract

Disgust associated with insect consumption is a significant challenge faced by the insect-based food industry. One cost-effective approach that managers can employ to increase consumer acceptance is by enhancing packaging design. The packaging represents a cheap and effective means of communication. It is also referred to as a silent seller. This study investigates the potential of packaging communication in reducing disgust towards insect-based products in Germany. In a survey, 422 participants were confronted with packaging designs representing different visual and informative elements. The results showed that images of familiar ingredients and transparent windows on the packaging are particularly effective in reducing disgust. The presence of the organic and specific Ento seals significantly increased the assumed food safety. Claims about protein content and sustainability were less effective. Cricket images had a significant impact on increasing disgust. Practical implications for managers who are seeking to address consumer resistance towards insect-based food products are discussed.

## 1. Introduction

In recent years, there has been a growing interest in entomophagy, a practice which involves the consumption of insects as a source of food. Despite the proven sustainability, environmental friendliness, and cost-effectiveness of insects as a source of high-quality protein when compared to meat, the acceptance of edible insects in Western societies remains low [1,2]. The disgust factor has been identified as the main barrier to such rejection [3,4,5,6], playing a significant role in the acceptance of edible insects [1,4,5,7,8,9,10,11]. Disgust refers to the negative emotional response and aversion to certain foods or substances and can be influenced by cultural and personal experiences. To encourage greater acceptance and consumption of edible insects in Western countries, it is essential to tackle and minimize disgust through targeted marketing strategies. Studies suggest that marketing communication can help to overcome this barrier and influence consumer behavior [3,8,12]. However, in the context of the edible insect sector, a significant number of companies operating in this industry are characterized by limited financial resources, particularly in terms of allocating budgets for extensive marketing campaigns. This situation has been primarily observed due to the prevalence of small-scale businesses and startups.

Previous research indicates that the disgust response towards insect-based food is triggered before individuals actually taste the product [13,14]. White et al. (2016) [15] describe that 90% of purchase decisions are made after the shopper has examined the product package. Following this idea, one possible approach to increasing the appeal and marketability of insect-based food products is to focus on packaging design [16]. By investing in packaging design, companies can make insect-based food products more appealing and accessible to a broader range of consumers, potentially increasing their sales and profitability. This strategy may also be cost-effective, as it may not require significant investments in marketing or advertising.

Recent studies have highlighted the pivotal role of packaging and communication in shaping consumer perceptions and acceptance of insect-based foods. Ref. [17], in their systematic review, showed that specific claims, for instance, regarding health and the environment, on packaged insects are beneficial in promoting the acceptance of alternative proteins, such as insect-based products. Ref. [18] also highlighted the positive influence of informative packaging on consumers’ willingness to pay for insect-based food products. Furthermore, the design of the packaging, such as its aesthetic appeal, can significantly affect consumers’ perception and expected acceptance of insect-based products [19]. Emphasizing the importance of design interventions [20] suggests that strategic food and packaging design can serve as an effective adoption strategy, enhancing consumer acceptance of edible insects. However, despite these insights, there remains a gap in understanding the optimal strategies for startups, especially given the high mortality rate of insect-based food startups in regions such as Europe [21]. Addressing this gap can provide actionable insights for businesses looking to penetrate the Western market with insect-based food products.

This research examines how disgust towards insect-based foods can be reduced through packaging communication, with the aim of providing insight into the factors that influence consumer acceptance. For the first time in the literature, we assess how the combination of labels (well-known organic labels and not widely known entomophagy-related labels), various claims and visual elements (pictures and transparent windows) can affect German consumers’ perception and acceptance of innovative insect-based products.

In the subsequent sections of this paper, we systematically unfold our research approach and findings, beginning with a review of the relevant literature presented as a theoretical model. Then, we delve into the methodology employed, shedding light on the data collection through an online survey and data analysis. This is followed by a presentation of the results and a discussion of the effectiveness of the packaging elements on disgust reduction textualized with previous studies. Finally, we present implications for managers and marketers and limitations and future directions.

### 1.1. Theoretical Model

#### 1.1.1. Disgust towards Entomophagy

Disgust for edible insects can be triggered by stimuli such as visual (appeal) aspects, negative taste expectations and food safety concerns. According to Deroy et al. (2015) [22], appeal aspects do often arouse disgust with insects. Insects that are not visible, such as ingredients in processed products, cause significantly less disgust than whole, unprocessed insects [4,7,8,9,23,24,25,26]. When insects are incorporated into familiar preparations (such as cookies or hamburgers) and preparations with an appetizing appearance, consumer acceptability can be enhanced. Negative taste expectations play an essential role in disgust [7,8]. Negative test expectations are a basic function of disgust to protect the body from consuming dangerous substances [27]. Finally, disgust is also associated with food safety concerns. Insects are considered a risk for disease transmission and are therefore perceived as dirty [28,29].

To measure the triggers of disgust (appeal aspects, taste expectations and food safety concerns), we tested the effect of informative (labels, claims and product information) and visual elements (pictures, imagery and transparent windows) according to the description of packaging design presented by [30]. Those elements have proven to have a significant influence on the consumers’ willingness to buy products [31].

#### 1.1.2. Informative Elements: Labels

Labels on the packaging are endorsed by external public and private institutions to provide reliable information on the products on which they are placed and influence consumers’ perceptions [32,33]. They act as a seal of approval and should credibly convey to the consumer that the product meets certain criteria that are standardized across food product categories. The certification may involve formal testing [33], showing that the products have been tested by experts and, therefore, are trustworthy [34]. Labels can increase the purchase probability [35]. Recently, there have been the first attempts by organizations and nutritionists to establish specific entomophagy labels [36,37]. However, it is questionable whether new and unfamiliar insect labels are suitable for enabling a transfer of trust. Samant and Seo (2016) [38] found that the purchase decision is positively influenced the most by labels that are known and understood by the consumer (H1: Packaging with informative elements such as labels better determines disgust in consumer responses than packaging without).

#### 1.1.3. Informative Elements: Claims

Claims on food packaging are text-based information that can help to outline the benefits connected to the product. Nutrition and health claims provide information about ingredients and their quality (e.g., ‘high in protein’, or ‘10% less fat’) [33]. Sensory claims summarize the descriptions that concern appearance, taste, aroma or texture. Currently, sustainability claims such as resource consumption or CO_2_ emissions are increasingly found on product packaging [39].

The effect of claims on consumer behavior has been studied previously [9,16,40]. Claims have also been found to be particularly effective with health and nutrition-conscious consumers [40,41,42]. Moreover, these elements attract more attention than nutritional information on packaging and can strongly influence purchasing decisions and product perception [40]. In the case of edible insects, there is a paradox. On one side, studies assert that claims that highlight health or ecological benefits to promote entomophagy could enhance consumer acceptance [5,43,44] and can positively influence the purchase decision [16]. Similarly, nutritional and palatability claims have been effective in increasing consumers’ willingness to try insect-based foods and raising expectations of taste [45]. Contrarily, other researchers have shown that statements about the environmental and nutritional benefits of insect-based foods have little influence on consumer behavior [8,9,12,46] (H2: Packaging with informative elements such as nutritional, sustainability and taste claims better determines disgust in consumer responses than packaging without).

#### 1.1.4. Visual Elements: Pictures

The imagery on food packaging can have a major influence on product perception, and products with pictures are preferred over products without any [30,47]. There is evidence indicating that packaging elements, including visual graphics [48,49], can significantly impact consumers’ behavior towards food products, influencing sensory perception and purchase intentions. Chrysochou and Grunert (2014) [50] found that images can have an even stronger influence on product perception than informative elements. Product images provide consumers with information about product quality [51]. According to Smith et al. (2015) [52], images of unprocessed food on product packaging can give the impression of a more natural taste, especially when photographs are used Instead of illustrations. However, the influence of the communication elements on the packaging in connection with entomophagy has not been extensively studied [53]. Kauppi and van der Schaar (2020) [16] suggested that the use of analogies on an insect-based protein bar packaging can be an impactful method for marketing insects. Marquis et al. (2023) [19] showed that “cute” visuals on insect-based bread and chips had a positive effect on reported emotions, expected product liking, tastiness and purchase intentions among young Colombian and French consumers. Similarly, [53] demonstrated that insect-based products are perceived as less disgusting when the packaging contains a less realistic insect image (H3: Packaging with visual elements such as pictures better determines disgust in consumer responses than packaging without).

#### 1.1.5. Visual Elements: Transparent Packaging Windows

Transparent windows in the packaging can influence product perception, such as the perceived quality, freshness and health of a product [54,55]. However, Sabri et al. (2020) [56] found that transparent packaging is only beneficial for products with low perceived quality risk and does not have much impact on products with high perceived quality risk. In their study, Kauppi and Schaar (2020) [16] found that consumers would prefer to see the insect-based product before the purchase, so transparent packaging windows might be helpful (H4: Packaging with visual elements such as transparent windows better determines disgust in consumer responses than packaging without).

## 2. Materials and Methods

### 2.1. Survey

An online questionnaire was designed and disseminated in the German language through social networks of the authors, students, colleagues and project partners. This social media included Facebook, Instagram and WhatsApp. In addition, special survey platforms (PollPool and SurveyCircle) were used, in which questionnaires from other users were answered, and these users answered a questionnaire in return. The sampling procedure was not probabilistic or snowball. The aim was to represent a cross-section of society with the participants as well as possible. People who generally do not like chocolate bars were excluded from the study (via a filter question) to ensure that the measured disgust is not triggered by an aversion towards the product category.

The questions were based on the evaluation of packages of chocolate insect-based bars used as product example. Initially, the respondents answered two filter questions about whether they are of legal age (above 18 years) and whether they like chocolate bars. In both cases, a “Yes” answer was a prerequisite for participating in the survey.

Based on the theoretical model, the next statements were related to one hypothesis as follows. H1: Packaging with informative elements such as labels better determines disgust in consumer responses than packaging without; H2: Packaging with informative elements such as nutritional, sustainability and taste claims better determines disgust in consumer responses than packaging without; H3: Packaging with visual elements such as pictures better determines disgust in consumer responses than packaging without and H4: Packaging with visual elements such as transparent windows better determines disgust in consumer responses than packaging without. Statements were linked to one of the three factors that trigger disgust towards insect-based food: (1) appeal aspects, (2) taste expectations and (3) food safety concerns. A question about the (4) purchase probability was added to evaluate an additional factor of consumer acceptance. To qualify these factors, participants were shown different insect chocolate bar packaging that was modified in their design (Table 1). To measure these four factors a seven-point Likert scale was employed because it reveals a good degree of description about the motif and effectively engages the logical reasoning of the participants [6]. On the scale, participants had the choice between (1) not appealing at all and very appealing, (2) disgusting and delicious, (3) not safe and very safe and (4) very unlikely I will buy it and very likely I will buy it.

The research was conducted as a quasi-experiment by using a one-group pre-test–post-test design. During this process, participants were asked about certain aspects. After introducing a stimulus, the participants were questioned again. If a significant difference in the responses was observed, it was attributed to the stimulus. In this case, the factors measured before applying the stimulus were the dependent variables ‘taste expectations’, ‘visual appeal’ and ‘expected food safety’. To further understand the influence of communication elements on the purchase decision, the likelihood of purchase was also assessed. Participants were initially shown an image of the packaging of an insect-based food product without the visible communication elements to be tested (referred to as the control packaging or external variable). This control packaging provided just enough information to inform participants that the product was insect-based, such as a chocolate bar with ground crickets. After measuring ‘taste expectations’, ‘visual appeal’, ‘expected food safety’ and ‘purchase probability’, participants were shown an image of packaging with an added communication element (serving as a stimulus or independent variable). Subsequently, feelings of disgust were measured again (Figure 1).

The communication elements tested included labels, claims, images and transparent packaging windows (Figure 2). Each element had its own separate packaging to ensure that any observed effects could be attributed to individual elements, eliminating the possibility that changes were due to a combination of elements. If a significant difference in expressed disgust between the control packaging and the packaging with the communication element was found, it was inferred that the change was induced by the communication element. Care was taken to ensure that respondents did not assume the same product was depicted in every question, as previously seen labels or claims might subconsciously influence perception. Therefore, it was clarified in the questionnaire that each displayed packaging contained a different product.

Afterwards, food neophobia was assessed based on a statement in which respondents chose the one that best suited them from among three options: “I often try new foods and am curious about new products”; “I am open to new foods, but it should be close to what I already know” and “I am not interested in eating foods that I do not know, especially if they contain ingredients that I have never eaten before”. According to these three individual options, participants were rated with a low, medium or high level of food neophobia. The variable was thus coded as follows: 1 = low, 2 = medium and 3 = high food neophobia. Therefore, the question of whether they have eaten insects before or not was asked. Finally, socio–demographic information was collected.

### 2.2. Data Analysis

Wilcoxon signed-rank tests were used for statistical analysis because the data presented a non-normal distribution. For each test package, the responses on appeal aversion, taste expectation, food safety and purchase probability were compared with the responses for the control packaging. A total of 36 Wilcoxon signed-rank tests were therefore carried out. When one of the influencing factors of disgust was rated significantly higher, it was assumed that the tested communication element on the packaging was feasible to reduce disgust. By comparing the means of the answers, it was determined how much the level of disgust indicated for the test packaging differed from the level indicated for the control packaging. In order to make statements about the effect size of the individual packaging communication elements in the acceptance, a Pearson correlation coefficient was calculated.

### 2.3. Sample Description

Four hundred twenty-two responses were received. Participants’ mean age was 38 ± 7.2 and ranged between 18 and 78 years. The sample was represented by 49 and 43% of males and females, respectively. A total of 7.7% belonged to diverse groups or did not respond. Two participants stated that they did not like chocolate bars and were therefore excluded from the evaluation.

## 3. Results

For a large part of the respondents, edible insects are still something new and unfamiliar, with 83% of the participants stating that they had never tried insect-based food before, and 17% had tried it once or several times. The share of participants with a high level of food neophobia (“I have no interest in trying foods that are unfamiliar to me”) was 2%, which is rather low. The majority (83%) chose the option “I like to try new foods if they are similar to familiar products”, which categorizes them as medium food neophobic. A total of 15% stated that they like to try new foods, even if they are completely novel to them, and fall under the group of a low level of food neophobia.

### Effectiveness of the Packaging Elements on Disgust Reduction

When the different packaging was compared with the control, average appeal aspects (from not appealing at all to very appealing) were significantly higher for packaging with a serving suggestion, chocolate bar and non-insect (I: x^−^ = 5.5 ± 0.5, z = −14.9, *p* < 0.00); followed by packaging with a transparent window (J: x^−^ = 4.9 ± 0.5 z = −17.4, *p* < 0.00); packaging with a serving suggestion, chocolate bar and insect (H: x^−^ = 3.4 ± 0.5 z = −13.9, *p* < 0.00); and with a taste claim (F: x^−^ = 2.8 ± 0.6 z = −7.2, *p* < 0.00). Packaging with an organic label (B: z = −13.9, *p* < 0.00), Entotrust label (C: z = −3.9, *p* < 0.00) and sustainability claim (E: z = −3.3, *p* < 0.00) obtained, on average, the same score (x^−^ = 2.6 ± 0.6). The lowest significant score was observed for packaging with a cricket image (G: x^−^ = 1.5 ± 0.5, z = −17.3, *p* < 0.00). No significative differences were observed when the packaging contained a nutritional claim (x^−^ = 2.5 ± 0.5, D: z = −2.2, *p* = 0.03). The appealing aspect was most influenced by the packaging with a serving suggestion, chocolate bar and non-insect (H: r = −0.85), followed by the packaging with a transparent window (J: r = −0.85) (Table 2).

On average, taste expectations (from disgusting to delicious) were significantly higher when compared with the control for packaging with a serving suggestion, chocolate bar and non-insect- (I: x^−^ = 5.5 ± 0.5, z = −15.4, *p* < 0.00;), followed by packaging with a transparent window (J: x^−^ = 5.0 ± 0.5, z = −17.4, *p* < 0.00); with a taste claim (F: x^−^ = 4.0 ± 0.5, z = −15.9, *p* < 0.00); and with a serving suggestion, chocolate bar and insect (H: x^−^ = 3.9 ± 0.5, z = −14.9, *p* < 0.00). The lowest significant score for taste expectations was observed for packaging with a cricket image (G: x^−^ = 1.7 ± 0.6; z = −16.5, *p* = 0.31). For packaging with the organic label (B: x^−^ = 2.7 ± 0.6, z = −1.4, *p* = 0.15), Entotrust label (C: x^−^ = 2.7 ± 0.6, z = −1.0, *p* = 0.31), containing a nutritional claim (D: x^−^ = 2.6 ± 0.5, z = −1.7, *p* = 0.11) and a sustainability claim (E: x^−^ = 2.6 ± 0.6, z = −1.4, *p* = 0.15), no significant differences were observed. The taste expectations were most influenced by the packaging with a transparent window (J: r = −0.82), followed by the packaging with a serving suggestion, chocolate bar and both non-insect and insect (J: r = −0.85 and H: r = −0.73, respectively).

In terms of perceived food safety (from not safe to very safe), the highest significant score was observed for the packaging with the organic and Entotrust labels (B: x^−^ = 6.4 ± 0.6, z = −15.9, *p* < 0.00 and C: x^−^ = 5.8 ± 0.5, z = −15.4, *p* < 0.00, respectively). It was followed by the packaging with a serving suggestion, chocolate bar and non-insect (I: x^−^ = 5.6 ± 0.5, z = −14.9, *p* < 0.00); with a transparent window (J: x^−^ = 5.2 ± 0.6, z = −14.3, *p* = < 0.00); with a taste claim (F: x^−^ = 4.3 ± 0.6, z = −3.2, *p* < 0.00); and with a serving suggestion, chocolate bar and insect (H: x^−^ = 4.3 ± 0.6, z = −3.1, *p* < 0.00). No significant differences were observed for the packaging containing a nutritional claim (D: x^−^ = 4.2 ± 0.6, z = −1.6, *p* = 0.09) and a sustainability claim (E: x^−^ = 4.2 ± 0.6, z = −1.4, *p* < 0.15). The perceived food safety was most influenced by the packaging with the organic label (B: r = −0.78), followed by the packaging with a serving suggestion, chocolate bar and non-insect (I: r = −0.72); Entotrust label (C: r = −0.75); and packaging with a transparent window (J: r = −0.70).

Regarding purchase probability, all packaging was significantly different from the control. The packaging that had the highest intention to be purchased was with a transparent window (J: x^−^ = 4.6 ± 0.6, z = −17.8, *p* < 0.00) and the packaging with a serving suggestion, chocolate bar and non-insect (I: x^−^ = 4.6 ± 0.6, z = −17.1, *p* < 0.00), followed by a serving suggestion, chocolate bar and insect (H: x^−^ = 3.1 ± 0.6, z = −14.9, *p* < 0.00); with a taste claim (F: x^−^ = 3.1 ± 0.5, z = −15.4, *p* < 0.00); and with a nutritional claim (D: x^−^ = 2.7 ± 0.5, z = −9.8, *p* < 0.00). Packaging with organic and Entotrust labels and a sustainability claim (B: z = −4.3, *p* < 0.00; C: z = −5.2, *p* < 0.00, respectively, and E: z = −3.2, *p* < 0.00) obtained, on average, the same scores (x^−^ = 2.4 ± 0.6). The purchase probability was most influenced by the packaging with a serving suggestion, chocolate bar and non-insect; with a transparent window (I and J: r = −0.84); and with a taste claim (F: r = −0.80).

## 4. Discussion

The present research offers relevant contributions to managers and marketers who seek economic and innovative strategies to improve consumer acceptance towards insect-based food acceptance. Our results can be implemented as part of retailers’ marketing strategy to alleviate the disgust that is generated from insect products, even before consumption. To the best of our knowledge, our study is the first that examines how the combination of labels, (well-known organic labels and not widely known entomophagy-related labels), various claims (taste, nutritional and sustainability) and visual elements (pictures and transparent windows) can affect German consumers’ perception and acceptance of insect-based products. Furthermore, we introduce a prototype packaging that offers initial empirical evidence validating the efficacy of marketing interventions designed to decrease consumers’ perceived disgust and enhance their willingness to try.

### 4.1. Effect of Labels on Disgust

The appeal aspect, food safety perception and purchase probability have been influenced by the tested labels when contrasted with the control packaging. Especially, the food safety factor had a positive influence on acceptance, which indicated the presence of a “halo” effect [58]. On one hand, it might have to do with the idea that labels are designed as informative elements that the consumer can trust [59,60,61]. On the other hand, it has been suggested that consumers place greater value on scientific testing labels when it comes to ensuring the quality and safety of food [60]. In addition, food safety perception has been described as an important determinant in consumers’ willingness to eat insects in Western countries, as reviewed by. [62]. Our results showed that the German organic label and the new Entotrust label are highly suitable for increasing perceived food safety with insect-based chocolate bars, confirming H1. Whereas the organic label is known as being associated with trust in the control of the production process, the new Entotrust label might profit from a kind of spillover effect from one label to another [29,30]. For example, well-known labels have been proven to impact consumers’ behavior towards insect-based food. Kuff et al. (2023) [63] showed that country-of-origin labeling (COOL) had a positive effect on both intentions to consume and quality expectations of cricket flour. However, the effect on the appeal aspect, food safety perception and purchase probability of the labels did not extend to taste expectations. Surprisingly, label packaging was not rated as significantly different from the control packaging on taste expectations. As suggested by Ellison et al. (2016) [58], the lack of perception of taste differences may be an influential factor that ultimately deters consumers from purchasing the product despite the organic halo effect. In order to develop more effective marketing strategies that can encourage the purchase of insect-based foods, a deeper understanding of the factors that influence consumer perceptions, such as cost and taste perception, should be further studied in the future.

### 4.2. Effect of Claims on Disgust

Studies have suggested that claims such as nutrition (high in protein) and the environment (protects resources) are the unique selling points of insects as novel food ingredients [64] and increase consumer acceptance [11,65,66,67]. Health concerns and a desire for sustainability are major factors that influence modern food choices. It has been proven that the health factor is a key driver of food selection [68,69], together with a focus on the natural environment and sustainability [70]. Contrary to previous studies cited, our data suggest that those claims have a low effect in reducing disgust towards the insect chocolate bar. We observed a significant increase in the appeal aspects of the packaging with the sustainability claim but not with the nutritional claim. The only measure that increased for both nutritional and sustainability claims was purchase probability, such as in the study conducted by 19 where environment claims had a significant effect on purchase intentions. Lombardi et al. (2019) [71] observed that when information on the benefits of insect consumption is provided (including nutritional, environmental and food safety) the consumers’ willingness to pay for insect-based products is increased. However, there is still limited evidence of significant effects resulting from the provision of supplementary information about the sustainability and nutritional benefits of insect-based food on consumer acceptance [8,9,12]; therefore, this topic should be studied in more depth to determine its relevance to the insect-based food sector. On the other hand, the claim related to the taste increased significantly in all the measures. Kauppi and Schaar (2020) [16] stated that providing information about the good taste of insects can positively influence acceptance. Based on our findings, there is evidence to suggest that informative elements on packaging can have an impact on consumer disgust responses. However, this influence is not absolute, and there are instances where packaging without these elements also elicited strong disgust responses. Therefore, while our results provide some support for H2, it is only partially accepted. Our data provide empirical evidence in favor of the notion that to mitigate the aversion commonly associated with entomophagy, marketing efforts should prioritize highlighting the hedonic properties of insect-based food items over emphasizing their sustainability and protein content.

### 4.3. Effect of Pictures on Level of Disgust

As hypothesized (H0), the participants found the control packaging visually unappealing and the taste experience of it rather negative. The lack of colors, structural elements and pictures in the control packaging could negatively affect product perception [72]. Pictures and graphics could be a potent tool to either increase or decrease disgust. Previous studies have suggested avoiding showing real insects on food packages to not trigger consumers’ disgust, and consumers in Western countries may experience negative emotions when confronted with the image of whole insects [19,23,53,73,74]. In general, they prefer to see an abstract image of an insect or “cute” designs on the packaging [16,23,53]. Similarly, we observed that associations with an image of a real insect seem to be overwhelmingly negative for German consumers. The image of a cricket increased all parameters of disgust and lowered the (already low-rated control package) probability of purchase. Nevertheless, we observed that when the packaging presented an image of a real insect, together with popular ingredients such as chocolate pieces and hazelnuts, the appeal and taste expectations were significantly positively influenced. Through the utilization of images depicting recognizable food ingredients, it is plausible that the product became linked with a familiar taste profile, thereby facilitating the mitigation of disgust even in response to realistic images. This indicates that while certain visual elements can enhance consumer responses, others, like the insect image, can amplify feelings of disgust. Given these results, we accept H3. The impact of visual elements on consumer disgust responses is evidently nuanced and varies depending on the specific imagery used. It has been previously suggested that pictures can have an even stronger influence on product perception than informative communication elements [50]. Baker and Shin (2016) [31] proposed that insect products should be advertised more prominently with pictures instead of additional information in a retail environment, as people tend to minimize their shopping time and pictures can be processed faster than text.

## 5. Conclusions

This research delves into the study of product packaging strategies for insect-based foods, aiming to uncover methods that can enhance consumer acceptance and minimize the disgust factor. Through our research, we have identified practical recommendations for the design and communication elements of packaging. These insights are not only pivotal for the successful marketing of insect-based food products but also for understanding the broader dynamics of consumer behavior in the face of novel food products.

### 5.1. Implications for Managers and Marketers

Our study uncovered significant implications for managers and marketers regarding the product packaging strategies of insect-based foods. One crucial implication for enhancing consumer acceptance and minimizing disgust is to incorporate product pictures showcasing familiar ingredients that evoke positive taste associations. Our findings indicated that most consumers are more likely to try new foods if they bear a resemblance to familiar ingredients. From a practical standpoint, marketers could incorporate food neophobia as an additional factor for consumer segmentation, as individuals may demonstrate varying levels of aversion towards specific foods or ingredients [75].

As a second implication, claims about taste should also be displayed on the packaging. Associations with already known flavors or ingredients can be helpful to create positive taste expectations and reduce disgust [26]. On the other hand, claims related to sustainability and protein content can highlight the benefits and added value of the product; however, they are not very effective in reducing disgust. According to earlier research, promoting the nutritional and environmental benefits of edible insects may not be effective in enhancing consumers’ willingness to try them, particularly if they do not expect their taste [8,46]. The sensory aspects of insect-based food products are essential in facilitating their integration into consumers’ daily dietary practices [75].

The third implication is that images of real insects should be avoided, as they can intensify disgust [16,19,53]. If associations with insects need to be made, an abstract form should be used instead. According to Koch et al. (2021) [14], employing less realistic images of insects may temporarily alleviate disgust reactions while concurrently increasing the public’s visibility of insect-based foods, thereby reducing their perceived novelty.

The fourth implication is that transparency on the packaging is also crucial to building trust and increasing the perceived food safety of the product [16]. This can be achieved by incorporating a transparent window in the packaging. If this is not feasible, product pictures can be used as an alternative. It is important to note that both transparency and product pictures have a positive effect on taste expectations, visual appeal and purchase probability. As a final implication, labels and certifications are relevant elements to building trust and increasing perceived food safety [62,76]. Organic labels, such as the bio label, and entomophagy-specific labels, such as Entotrust, are highly effective. However, if the financial resources are available and the standards for a well-known certificate can be met, it is recommended to use well-known labels, as they are slightly more effective than an unknown label.

These recommendations need to be aligned with the corporate strategy and corporate design to ensure successful implementation. Not all suggestions need to be implemented, as some products may not cause significant disgust. Before launching a new product, companies should conduct surveys to identify the areas where disgust needs to be reduced. Based on those implications, we elaborate on a packaging prototype as a suggestion for an insect chocolate bar as an example (Figure 3). Overall, managers should focus on utilizing informative labels and claims while being cognizant of the visual elements used on the packaging to effectively market insect-based food products and increase their appeal among consumers.

### 5.2. Limitations and Future Directions

Although our research generated practical knowledge for the marketing sector of edible insects, it has limitations that could be explored in future research. This work is limited to general recommendations for the design of product packaging. It is not meant to advise on specific product categories but could be related to the market for sweets with insect ingredients. Here, we only examine products in which insects are processed and not products with whole, visible insects, as these potentially meet with the highest acceptance for German consumers [4,9,77].

We tested different elements of packaging communication on a general group of consumers; however, future research on identifying specific segments and determining how packaging design can be tailored to effectively target these segments is still necessary. Pozharliev et al. (2023) [75] proposed that levels of neophobia related to the presentation of images in the package and health consciousness measures should be used as criteria for segmentation. This could involve understanding the preferences and motivations of different consumer groups and developing packaging strategies that align with their specific needs and values.

Future research should be extended beyond online surveys to examine the impact of packaging communication on reducing disgust in real-world purchasing scenarios. Our prototype (or modified, according to particular needs) could be used to conduct field experiments or observational studies in supermarkets or other retail environments to assess how consumers react to insect-based food packaging and whether it influences their purchasing decisions. It should also be mentioned that disgust is not the only reason for the rejection of insect-based foods since factors such as price also play a role [71]. This factor should be measured as well in field experiments.

Finally, in our study, we have specifically evaluated a German population. Cross-cultural studies have identified that there are differences regarding insect-based food acceptance between consumers from different countries [7,8,78]. Further research could investigate variations of our (modified) prototype across different cultural backgrounds.

## Figures and Tables

**Figure 1 foods-12-03606-f001:**
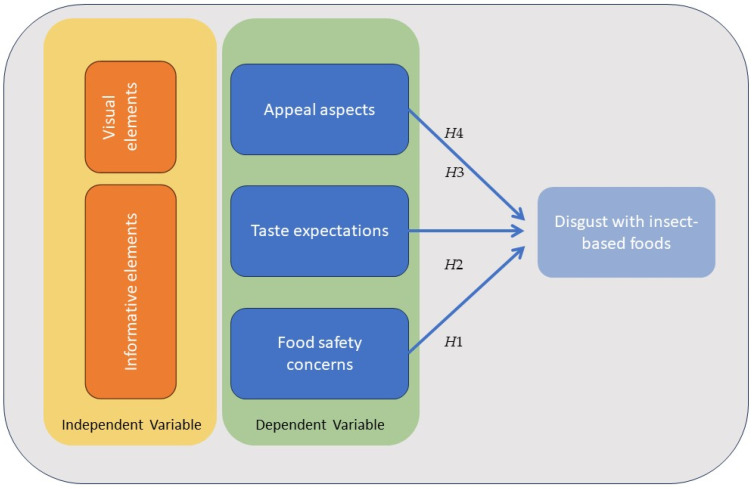
Schematic representation of the quasi-experimental one-group pre-test–post-test design and hypothesis model. The impact of added communication elements on packaging (independent variable) on participants’ taste expectations, visual appeal, expected food safety and purchase probability (dependent variables). H1: Packaging with informative elements such as labels better determines disgust in consumer responses than packaging without; H2: Packaging with informative elements such as nutritional, sustainability and taste claims better determines disgust in consumer responses than packaging without; H3: Packaging with visual elements such as pictures better determines disgust in consumer responses than packaging without and H4: Packaging with visual elements such as transparent windows better determines disgust in consumer responses than packaging without.

**Figure 2 foods-12-03606-f002:**
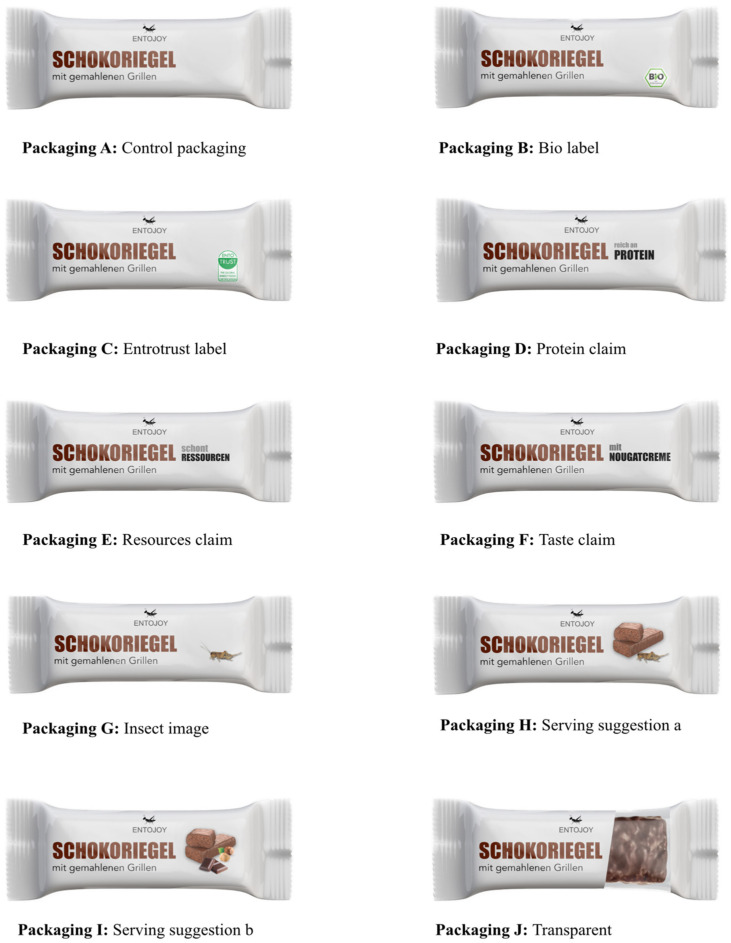
Control insect chocolate bar package (**A**) and its modifications (**B**–**J**), allowing testing for the influence of the factors that trigger disgust.

**Figure 3 foods-12-03606-f003:**
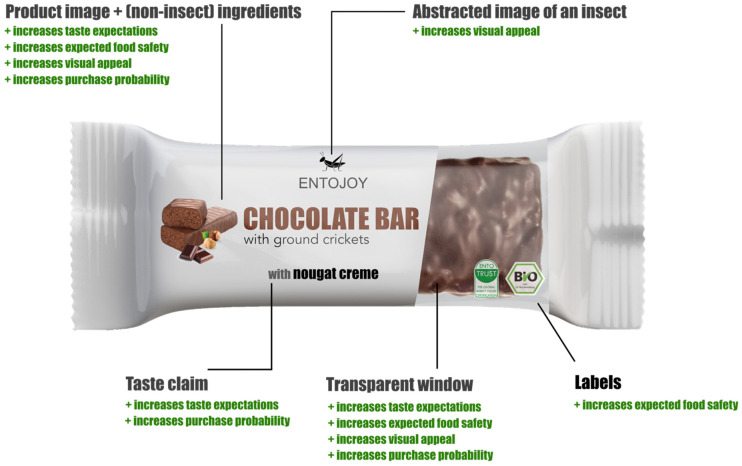
Packaging prototype and effect of different elements of visual communication.

**Table 1 foods-12-03606-t001:** Relationship between hypotheses, tested elements and the description of the package.

Hypothesis	Informative/Visual Elements	Description	Packaging
	Control packaging	Neutral without informative or visual elements	A
H1	Packaging with an organic label	One of the best-known labels in Germany [57]	B
	Packaging with the Entotrust label	New and thus unknown entomophagy-related label of Entotrust [36]	C
H2:	Packaging with a nutritional claim	“High in protein” (*reich an Protein*)	D
	Packaging with a sustainability claim	“Protects resources” (*schont Ressourcen*)	E
	Packaging with a taste claim	“With nougat cream” (*mit Nougatcreme*)	F
H3	Packaging with a cricket image	A picture of a cricket	G
	Packaging with a serving suggestion (I)	Picture of insect and chocolate bar	H
	Packaging with a serving suggestion (II)	Picture of chocolate bar and non-insect ingredients	I
H4:	Packaging with transparent window	Showing a part of the chocolate bar	J

**Table 2 foods-12-03606-t002:** Effect of informative and visual elements of an insect bar on consumer perception.

Packaging	Appeal Aspects	*p*	r	Taste Expectations	*p*	r	Food Safety	*p*	r	Purchase Probability	*p*	r
A	2.5 ± 0.6	n.c	n.c	2.6 ± 0.6	n.c	n.c	4.2 ± 0.7	n.c	n.c	2.3 ± 0.5	n.c	n.c
B	2.6 ±0.5	<0.00	−0.2	2.7 ± 0.6	0.15	−0.1	6.4 ± 0.6	<0.00	−0.8	2.4 ± 0.6	<0.00	−0.2
C	2.6 ± 0.6	<0.00	−0.2	2.7 ± 0.6	0.31	n.c	5.8 ± 0.5	<0.00	−0.7	2.4 ± 0.6	<0.00	−0.2
D	2.5 ± 0.5	0.03	n.c	2.6 ± 0.5	0.11	n.c	4.2 ± 0.6	0.09	n.c	2.7 ± 0.5	<0.00	−0.2
E	2.6 ± 0.6	<0.00	−0.2	2.6 ± 0.6	0.15	n.c	4.2 ± 0.6	0.15	n.c	2.4 ± 0.6	<0.01	−0.2
F	2.8 ± 0.6	<0.00	−0.4	4.0 ± 0.5	<0.00	−0.8	4.3 ± 0.6	<0.00	−0.2	3.1 ± 0.5	<0.00	−0.8
G	1.5 ± 0.5	<0.00	n.c	1.7 ± 0.6	<0.00	n.c	3.3 ± 0.7	<0.00	n.c	1.5 ± 0.5	<0.00	n.c
H	3.4 ± 0.5	<0.00	0.6	3.9 ± 0.5	<0.00	−0.7	4.3 ± 0.6	<0.00	−0.1	3.1 ± 0.6	<0.00	0.7
I	5.5 ± 0.5	<0.00	−0.8	5.5 ± 0.5	<0.00	−0.8	5.6 ± 0.5	<0.00	0.7	4.6 ± 0.6	<0.00	−0.8
J	4.9 ± 0.5	<0.00	−0.8	5.0 ± 0.5	<0.00	−0.8	5.2 ± 0.6	<0.00	−0.7	4.6 ± 0.6	<0.00	−0.8

n.c: significant difference at *p* < 0.05 Wilcoxon signed-rank tests values.

## Data Availability

The data presented in this study are available on request from the corresponding author.

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
