# Peer review of "Packaging Communication as a Tool to Reduce Disgust with Insect-Based Foods: Effect of Informative and Visual Elements"

_foods, 2023, doi:10.3390/foods12193606_

Round 1
Reviewer 1 Report
The authors present an experimental study on the effect of packaging information in insect-based food products, aimed to assess consumer disgust in product consumption.
Paper writing and conceptualization are in line with the scope of the journal. Bibliographic sources throughout the paper appear adequate, and the topic is of interest to the scientific community and justifies the purpose of the research. Overall scientific content of the paper shows a correct methodological approach, both in the choice of survey methodology and testing.
The results of the survey led to the achievement of the key parameters for maximizing purchase propension for the food product. Also, the results of the survey are effectively represented by the prototype of the packaging produced that clearly convey the idea of the best possible combination of elements leading to product purchase.
Minor text and spelling issues should be corrected.
English language fine. Few minor text and spelling issues.
Author Response
Dear reviewer,
Text and spelling issues were corrected.
We appreciate your valuable feedback!

Reviewer 2 Report
The primary purpose of this report is to give an overview of the paper entitled "Packaging Communication as a Tool to Reduce Disgust with Insect-based Foods: Effect of Informative and Visual Elements". No doubt, the paper brings up the issue of an important topic and, overall, it deserves this reviewer's care. Therefore, it can be stated that the paper is bound to get publication if the authors pay careful attention to the following suggestions:
(1) Although the authors find grounds to deal with this subject matter (lines 23-40) and set out brilliantly general and specific research objectives (lines 50-60), the authors do not spot the research gap from a purely scientific point of view. Please address the research gap by referring to the research variables and citing other papers with JCR impact.
(2) Although the paper structure is classic and straightforward, it would be advisable to append an advance of its structure to the introduction. Please write a paragraph at the end of the introduction to describe briefly the paper’s structure.
(3) The literature review section should deal with theoretical content rather than methodological content. However, the paragraph between lines 74-79 looks quite methodological, as it tackles measuring instruments. Please revise it, move it to the methodology section or delete it.
(4) The literature review section lacks hypotheses even though it seems close to putting forward them. For example, “H1: Packaging with label informative elements determine more/less disgust consumer responses than packaging without” might be inserted in line 93. Similar hypotheses regarding a claim with informative elements (line 113), pictures with visual elements (line 133) and visual elements with transparent packaging windows (line 142).
(5) The methodology needs more crucial information. First, the authors should inform the reader about the survey context regarding when, where and who performed the survey. What survey platforms and social media are you referring to (line 146)? Second, the measuring instrument should be inspired by the literature, and hence, the scales should be based on bibliographical references (lines 155-160). Third, there needs to be more information about the sampling procedure. Was it probabilistic or not probabilistic? Was it convenient or snowball?
(6) Although the research is based on a survey, it might be considered a quasi-experiment because the different types of packaging might be labelled as independent variables. The authors are measuring the effect of the independent variables on the dependent variables, and hence, it might be described as a kind of experiment. However, it is hard to identify the independent and dependent variables, and nothing is said about the potential influence of external variables. Therefore, the authors should describe it as a kind of quasi-experiment by indicating the independent, dependent and external variables, the treatments and the experimental subject units. Moreover, I recommend that the authors create and design a figure to represent the quasi-experiment design visually. Please describe and visualise the experiment in the methodology.
(7) Table 1 displays information about six alternative hypotheses and no null hypothesis. Nevertheless, the assumed hypotheses are not stated; hence, they remain in the dark. Moreover, putting forward more than one null hypothesis must make sense, having formulated six alternative hypotheses. Please delete this misleading information.
(8) The Wilcoxon test is a non-parametric statistical test; hence, it could be used with dependent samples to compare two different samples. Simply put, it is a difference test (lines 180-188). Conversely, the correlation test is a similarity test; hence, it is not suitable to measure the differences between samples but rather the similarities. The correlation test is a statistical tool to measure similarities and a descriptive statistical tool whose technical features can not be causal (lines 189-190). By the way, it does make sense to talk about the correlation test in the methodology if it is not used to analyse the result. Is it used? Please pay careful attention to these comments and proceed conveniently.
(9) The sample description is not suitable for the results section (lines 191-197) but rather the methodology. Other descriptive data might be left in the result section (lines 199-207). Please move the description of the sample to the results section.
(10)The discussion section is insightful, but it encompasses unsuitable content such as a summary (lines 263-273), practical implications (lines 347-393), limitations and future lines of research (lines 394-422). As far as a conclusion section is concerned, a summary, practical implications, limitations and future lines of research should be developed. Please create a conclusion section to move these paragraphs.
I hope these comments help improve the paper and encourages the authors to move forward
Author Response
Dear reviewer,
Please find the answers to your suggestions below:
- Although the authors find grounds to deal with this subject matter (lines 23-40) and set out brilliantly general and specific research objectives (lines 50-60), the authors do not spot the research gap from a purely scientific point of view. Please address the research gap by referring to the research variables and citing other papers with JCR impact.
We have added a new paragraph between lines 51 and 65 in which we explain the research gap considering literature from indexed academic journals.
- Although the paper structure is classic and straightforward, it would be advisable to append an advance of its structure to the introduction. Please write a paragraph at the end of the introduction to describe briefly the paper’s structure.
We have added a new paragraph between lines 74-80
- The literature review section should deal with theoretical content rather than methodological content. However, the paragraph between lines 74-79 looks quite methodological, as it tackles measuring instruments. Please revise it, move it to the methodology section or delete it.
We moved this part to the methodology.
- The literature review section lacks hypotheses even though it seems close to putting forward them. For example, “H1: Packaging with label informative elements determine more/less disgust consumer responses than packaging without” might be inserted in line 93. Similar hypotheses regarding a claim with informative elements (line 113), pictures with visual elements (line 133) and visual elements with transparent packaging windows (line 142).
Thanks for the suggestion. We have added text about the hypotheses at the end of each section as suggested.
- The methodology needs more crucial information. First, the authors should inform the reader about the survey context regarding when, where and who performed the survey. What survey platforms and social media are you referring to (line 146)? Second, the measuring instrument should be inspired by the literature, and hence, the scales should be based on bibliographical references (lines 155-160). Third, there needs to be more information about the sampling procedure. Was it probabilistic or not probabilistic? Was it convenient or snowball?
We have added text about the methodology and Likert point scale as suggested (Lines 176-204)
- Although the research is based on a survey, it might be considered a quasi-experiment because the different types of packaging might be labelled as independent variables. The authors are measuring the effect of the independent variables on the dependent variables, and hence, it might be described as a kind of experiment. However, it is hard to identify the independent and dependent variables, and nothing is said about the potential influence of external variables. Therefore, the authors should describe it as a kind of quasi-experiment by indicating the independent, dependent and external variables, the treatments and the experimental subject units. Moreover, I recommend that the authors create and design a figure to represent the quasi-experiment design visually. Please describe and visualise the experiment in the methodology.
We are grateful for the recommendation provided. We have incorporated a section elucidating the quasi-experimental design, delineating the variables, and detailing the execution of the stimuli. It is our aspiration that this elaboration sufficiently clarifies the methodology, obviating the need for an additional figure (lines 209-236)
- Table 1 displays information about six alternative hypotheses and no null hypothesis. Nevertheless, the assumed hypotheses are not stated; hence, they remain in the dark. Moreover, putting forward more than one null hypothesis must make sense, having formulated six alternative hypotheses. Please delete this misleading information.
We have added text about the hypothesis in the table 1. We have deleted the misleading information as well.
- The Wilcoxon test is a non-parametric statistical test; hence, it could be used with dependent samples to compare two different samples. Simply put, it is a difference test (lines 180-188). Conversely, the correlation test is a similarity test; hence, it is not suitable to measure the differences between samples but rather the similarities. The correlation test is a statistical tool to measure similarities and a descriptive statistical tool whose technical features can not be causal (lines 189-190). By the way, it does make sense to talk about the correlation test in the methodology if it is not used to analyse the result. Is it used? Please pay careful attention to these comments and proceed conveniently.
We include an explanation of the use of the Wilcoxon test as a non-parametric test.
The correlation test is presented in the results with a letter r and analysed at the end of each paragraph
- The sample description is not suitable for the results section (lines 191-197) but rather the methodology. Other descriptive data might be left in the result section (lines 199-207). Please move the description of the sample to the results section.
The paragraph was movede as suggested
(10)The discussion section is insightful, but it encompasses unsuitable content such as a summary (lines 263-273), practical implications (lines 347-393), limitations and future lines of research (lines 394-422). As far as a conclusion section is concerned, a summary, practical implications, limitations and future lines of research should be developed. Please create a conclusion section to move these paragraphs.
The conclusion section was created according to the suggestions (from line 424)
We appreciate the valuable feedback!

Round 2
Reviewer 2 Report
Please see the attachment.

Author Response
Thank you for your revise. Attachment is the response reply.
